# VARIATIONAL TEMPLATE MACHINE FOR DATA-TO-TEXT GENERATION

**Rong Ye**[†][*]**, Wenxian Shi, Hao Zhou, Zhongyu Wei**[†]**, Lei Li**
[†]Fudan University
`{rye18,zywei}@fudan.edu.cn`
ByteDance AI Lab
`{shiwenxian,zhouhao.nlp,lileilab}@.bytedance.com`

## ABSTRACT

How to generate descriptions from structured data organized in tables? Existing approaches using neural encoder-decoder models often suffer from lacking diversity. We claim that an open set of templates is crucial for enriching the phrase constructions and realizing varied generations. Learning such templates is prohibitive since it often requires a large paired `<table,description>` corpus, which is seldom available. This paper explores the problem of automatically learning reusable "templates" from paired and non-paired data. We propose the *variational template machine* (VTM), a novel method to generate text descriptions from data tables. Our contributions include: *a)* we carefully devise a specific model architecture and losses to explicitly disentangle text template and semantic content information in the latent spaces, and *b)* we utilize both small parallel data and large raw text without aligned tables to enrich the template learning. Experiments on datasets from a variety of different domains show that VTM is able to generate more diversely while keeping a good fluency and quality.

## 1 INTRODUCTION

Generating text descriptions from structured data (data-to-text) is an important task with many practical applications. Data-to-text has been used to generate different kinds of texts, such as weather reports (Angeli et al., 2010), sports news (Mei et al., 2016; Wiseman et al., 2017) and biographies (Lebret et al., 2016; Wang et al., 2018b; Chisholm et al., 2017). Figure 1 gives an example of data-to-text task, which takes an infobox [1] as the input and outputs a brief description of the information in the table. There are several recent methods utilizing neural encoder-decoder frameworks to generate text description from data tables (Lebret et al., 2016; Bao et al., 2018; Chisholm et al., 2017; Liu et al., 2018).

Although current table-to-text models could generate high quality sentences, the diversity of these output sentences are not satisfactory. We find that *templates* are crucial in increasing the variations of sentence structure. For example, Table 1 gives three descriptions with their templates for the given table input. Different templates control the sentence arrangement, thus vary the generation. Some related work (Wiseman et al., 2018; Dou et al., 2018) employs hidden semi-Markov hidden model to extract templates from table-text pairs.

We argue that templates can be better considered for generating more diverse outputs. First, it is non-trivial to sample different templates for obtaining different output utterances. Directly adopting variational auto-encoders (VAEs, Kingma & Welling (2014)) in table-to-text only enables to sample in the latent space. However, VAEs always generate irrelevant outputs, which may change the table content instead of sampling templates. This may harm the quality of output sentences. To address the above problem, if we can directly sample in the template space, we may get more diverse outputs while keeping the good quality of output sentences.

---

[*]Work done while Rong Ye was a research intern at ByteDance AI Lab.

[1]An infobox is a table containing attribute-value data about a certain subject. It is mostly used on Wikipedia pages.

| Table: | **name**[nameVariable], **eatType**[pub], **food**[Japanese], **priceRange**[average], **customerRating**[low], **area**[riverside] |
|---|---|
| **Template1**: | [name] is a [food] restaurant, it is a [eatType] and it has an [priceRange] cost and [customerRating] rating. it is in [area]. |
| **Sentence1**: | nameVariable is a Japanese restaurant, it is a pub and it has an average cost and low rating. it is in riverside. |
| **Template2**: | [name] has an [priceRange] price range with a [customerRating] rating, and [name] is an [food] [eatType] in [area]. |
| **Sentence2**: | nameVariable has an average price range with a low rating, and nameVariable is an Japanse pub in riverside. |
| **Template3**: | [name] is a [eatType] with a [customerRating] rating and [priceRange] cost, it is a [food] restaurant and [name] is in [area]. |
| **Sentence3**: | nameVariable is a pub with a low rating and average cost, it is a Japanese restaurant and nameVariable is in riverside. |

Table 1: An example: generating sentences based on different templates.

Second, we can hardly obtain promising sentences by sampling in the template space, if the template space is less informative. Namely, either encoder-decoder models or VAE-based models requires abundant parallel table-text pairs during the training. In such case, constructing high-quality parallel dataset is often labor-intensive. With limited table-sentence pairs, a VAE model cannot construct an informative template space. How to fully utilize raw sentences (without aligned table) to enrich the latent template space is under study.

In this paper, to address the above two problems, we propose the *variational template machine* (VTM) for data-to-text generation, which enables to generate sentences with diverse templates while preserving the high quality. Particularly, we introduce two latent variables, representing *template* and *content*, to control the generation. The two latent variables are disentangled, and thus we can generate diverse outputs by directly sampling in the latent space for template. Moreover, we propose a novel approach for semi-supervised learning in the VAE framework, which could fully exploit the raw sentences for enriching the template space. Inspired by back-translation (Sennrich et al., 2016; Burlot & Yvon, 2018; Artetxe et al., 2018), we design a variational back-translation process. Instead of training a sentence-to-table backward generation model directly, we take the variational posterior of the content latent variable as the backward model to help to train the forward generative model. Auxiliary losses are introduced to ensure the learning of meaningful and disentangled latent variables.

Experimental results on Wikipedia biography dataset (Lebret et al., 2016) and sentence planning NLG dataset (Reed et al., 2018) show that our model can generate texts with more diversity while keeping a good fluency. Training together with a large amount of raw text, VTM can further improve the generation performance. Besides, VTM is more predominant in the case where sentence-to-table backward model is hard to train. Ablation studies also demonstrate the effects of the auxiliary losses on the disentanglement of template and content spaces.

## 2 PROBLEM FORMULATION AND NOTATIONS

As a data-to-text task, we have **table-text pairs** $\mathcal{D}_p = \{(\boldsymbol{x}_i, \boldsymbol{y}_i)\}_{i=1}^N$, where $\boldsymbol{x}_i$ is the table, and $\boldsymbol{y}_i$ is the output sentence.

Following the description scheme of Lebret et al. (2016), a table $\boldsymbol{x}$ can be viewed as a set of $K$ records of field-position-value triples, i.e., $\boldsymbol{x} = \{(f, p, v)_i\}_{i=1}^K$, where $f$ is the field and $p$ is the index of value $v$ in the field $f$. For example, an item *"Name: John Lennon"* is denoted as two corresponding records: (*Name, 1, John*) and (*Name, 2, Lennon*). For each triple, we first embed field, position and value as $d$-dim vectors $e_p, e_f, e_v \in \mathbb{R}^d$. Then, the $d_t$-dim representation of the record is obtained by $h_i = \textbf{tanh}(W[e_f, e_p, e_v]^T + b)$, $i = 1...K$, where $W \in \mathbb{R}^{d_t \times 3d}$ and $b \in \mathbb{R}^{d_t}$ are parameters. The final representation of the table, denoted as $f_{enc}(x)$, is obtained by max-pooling over all field-position-value triple records,

$$f_{\text{enc}}(x) = h = \textbf{MaxPool}_i\{h_i; i = 1...K\}.$$

In addition to the table-text pairs, we also have **raw texts** without table input, denoted as $\mathcal{D}_r = \{\boldsymbol{y}_i\}_{i=1}^M$. It usually has $M \gg N$.

| Data Type | Structured Data (Source) | Descriptive Text (Target) |
|---|---|---|
| Table-text pairs | **Chris Larsen**
Born 1960 (age 58−59)[1] San Francisco, California
Nationality American
Education San Francisco State University (B.S.)
Alma mater Stanford Graduate School of Business (M.B.A.)
Occupation Angel investor, business executive
Years active 1990s−present
Employer Ripple Labs (Executive Chairman)
Net worth $4.6 billion | Chris Larsen (born 1960) is a business executive and angel investor best known for co-founding several Silicon Valley technology startups, including one based on peer to peer lending. |
| Raw text | Not provided | Mother Teresa (1910−1997) was a Roman Catholic nun who devoted her life to serving the poor and destitute around the world. |

Figure 1: Two types of data in the data-to-text task: Row 2 presents an example of table-text pairs; Row 3 shows a sample of raw text, whose table input is missing and only sentence is provided.

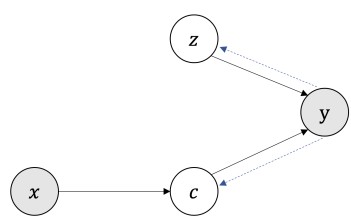

Figure 2: The graphical model of VTM: $z$ is the latent variable from template space, and $c$ is the content variable. $x$ is the corresponding table for the table-text pairs. $y$ is the observed sentence. The solid lines depict the generative model and the dashed lines form the inference model.

## 3 VARIATIONAL TEMPLATE MACHINE

As shown in the graphical model in Figure 2, our VTM modifies the vanilla VAE model by introducing two independent latent variables $z$ and $c$, representing **template** latent variable and **content** latent variable respectively. $c$ models the content information in the table, while $z$ models the sentence template information. Target sentence $y$ is generated by both content and template variables. The two latent variables are disentangled, which makes it possible to generate diverse and relevant sentences by sampling template variable and retraining the content variable. Considering pairwise and raw data presented in Figure 1, their generation process for the content latent variable $c$ is different.

- For a given table-text pair $(x, y) \in \mathcal{D}_p$, the content is observable from table $x$. As a result, $c$ is assumed to be deterministic given table $x$, whose prior is defined as a delta distribution $p(c|x) = \delta(c = f_{\text{enc}}(x))$. The marginal log-likelihood is:

$$
\begin{aligned}
\log p_\theta(y|x) &= \log \int_z \int_c p_\theta(y|x, z, c) p(z) p(c|x) \text{dcdz} \\
&= \log \int_z p_\theta(y|x, z, c = f_{\text{enc}}(x)) p(z) \text{dz}, (x, y) \in \mathcal{D}_p.
\end{aligned}
\tag{1}
$$

- For raw text $y \in \mathcal{D}_n$, the content is unobservable with the absence of table $x$. As a result, the content latent variable $c$ should be sampled from prior of Gaussian distribution $\mathcal{N}(0, I)$. The marginal log-likelihood is:

$$
\log p_\theta(y) = \log \int_z \int_c p_\theta(y|z, c) p(z) p(c) \text{dcdz}, y \in \mathcal{D}_r.
\tag{2}
$$

In order to make full use of both table-text pair data and raw text data, the above marginal log-likelihood should be optimized jointly:

$$
\mathcal{L}(\theta) = \mathbb{E}_{(x,y) \sim \mathcal{D}_p}[\log p_\theta(y|x)] + \mathbb{E}_{y \sim \mathcal{D}_r}[\log p_\theta(y)].
\tag{3}
$$

Directly optimizing Equation 3 is intractable. Following the idea of variational inference (Kingma & Welling, 2014), a variational posterior $q_\phi(\cdot)$ is constructed as an inference model (dashed lines in Figure 2) to approximate the true posterior. Instead of optimizing the marginal log-likelihood in Equation 3, we maximize the evidence lower bound (ELBO). In Section 3.1 and 3.2, the ELBO of table-text pairwise data and raw text data are discussed, respectively.

## 3.1 LEARNING FROM TABLE-TEXT PAIR DATA

In this section, we will show the learning loss of table-text pair data. According to the aforementioned assumption, the content variable $c$ is observable and follows a delta distribution centred in the hidden representation of the table $x$.

**ELBO objective.** Assuming that the template variable $z$ only relies on the template of target sentence, we introduce $q_\phi(z|y)$ as an approximation of the true posterior $p(z|y, c, x)$,

The ELBO loss of Equation 1 is written as

$$\mathcal{L}_{\text{ELBO}_p}(x, y) = -\mathbb{E}_{q_{\phi_z}(z|y)} \log p_\theta(y|z, c = f_{\text{enc}}(x), x) + D_{\text{KL}}(q_{\phi_z}(z|y)\|p(z)), \quad (x, y) \in \mathcal{D}_p.$$

The variational posterior $q_{\phi_z}(z|y)$ is assumed as a multivariate Gaussian distribution $\mathcal{N}(\mu_{\phi_z}(y), \Sigma_{\phi_z}(y))$, while the prior $p(z)$ is taken as a normal distribution $\mathcal{N}(0, I)$.

**Preserving-Template Loss.** Without any supervision, the ELBO loss alone does not guarantee to learn a good template representation space. Inspired by the work in style-transfer (Hu et al., 2017b; Shen et al., 2017; Bao et al., 2019; John et al., 2018), an auxiliary loss is introduced to embed the template information of sentences into template variable $z$.

With table, we are able to roughly align the tokens in sentence with the records in the table. By replacing these tokens with a special token $<ent>$, we can remove the content information from sentences and get the sketchy sentence template, denote as $\tilde{y}$. We introduce the preserving-template loss $\mathcal{L}_{\text{pt}}$ to ensure that the latent variable $z$ only contains the information of the template.

$$\mathcal{L}_{\text{pt}}(x, y, \tilde{y}) = -\mathbb{E}_{q_{\phi_z}(z|y)} \log p_\eta(\tilde{y}|z) = -\mathbb{E}_{q_{\phi_z}(z|y)} \sum_{t=1}^{m} \log p_\eta(\tilde{y}_t|z, \tilde{y}_{<t})$$

where $m$ is the length of the $\tilde{y}$, and $\eta$ denotes the parameters of the extra template generator. $\mathcal{L}_{\text{pt}}$ is trained via parallel data. In practice, due to the insufficient amount of parallel data, template generator $p_\eta$ may not be well-learned. However, experimental results show that this loss is sufficient to provide a guidance for learning a template space.

## 3.2 LEARNING FROM RAW TEXT DATA

Our model is able to make use of a large number of raw data without table since the content information of table could be obtained by the content latent variable.

**ELBO objective.** According to the definition of generative model in Equation 2, the ELBO of raw text data is

$$\log p_\theta(y) = \mathbb{E}_{q_\phi(z, c|y)} \log \frac{p_\theta(y, z, c)}{q_\phi(z, c|y)}, \quad y \in \mathcal{D}_r.$$

With the mean field approximation (Xing et al., 2003), $q_\phi(z, c|x)$ can be factorized as: $q_\phi(z, c|y) = q_{\phi_z}(z|y)q_{\phi_c}(c|y)$. We have:

$$\mathcal{L}_{\text{ELBO}_r}(y) = -\mathbb{E}_{q_{\phi_z}(z|y)q_{\phi_c}(c|y)} \log p_\theta(y|z, c)$$
$$+ D_{\text{KL}}(q_{\phi_z}(z|y)\|p(z)) + D_{\text{KL}}(q_{\phi_c}(c|y)\|p(c)), \quad y \in \mathcal{D}_r.$$

In order to make use of template information contained in raw text data effectively, the parameters of generation network $p_\theta(y|z, c)$ and posterior network $q_{\phi_z}(z|y)$ are shared for pairwise and raw data. In decoding process, for raw text data, we use content variable $c$ as the table embedding for the missing of table $x$. Variational posterior for $c$ is deployed as another multivariate Guassian $q_{\phi_c}(c|y) = \mathcal{N}(\mu_{\phi_c}(y), \Sigma_{\phi_c}(y))$. Both $p(z)$ and $p(c)$ are taken as normal distribution $\mathcal{N}(0, I)$.

**Preserving-Content Loss.** In order to make the posterior $q_{\phi_c}(c|y)$ correctly infers the content information, the table-text pairs are used as the supervision to train the recognition network of $q_{\phi_c}(c|y)$. To this end, we add a preserving-content loss

$$\mathcal{L}_{\text{pc}}(x, y) = -\mathbb{E}_{q_{\phi_c}(c|y)}\|c - h\|^2 + D_{\text{KL}}(q_{\phi_c}(c|y)\|p(c)), \quad (x, y) \in \mathcal{D}_p,$$

where $h = f_{\text{enc}}(x)$ is the embedding of table obtained by the table encoder. Minimizing $\mathcal{L}_{\text{pc}}$ is also helpful to bridge the gap of $c$ between pairwise (taking $c = h$) and raw training data (sampling from $q_\phi(c|y)$). Moreover, we find that the first term of $\mathcal{L}_{\text{pc}}$ is equivalent to (1) make the mean of $q_\phi(c|y)$ closer to $h$; (2) minimize the trace of co-variance of $q_\phi(c|y)$. The second term serves as a regularization. Detailed explanations and proof are referred in supplementary materials.

---

**Algorithm 1** Training procedure

---

**Input:** Model parameters $\phi_z, \phi_c, \theta, \eta$
    Table-text pair data $\mathcal{D}_p = \{(\boldsymbol{x}, \boldsymbol{y})_i\}_{i=1}^N$; raw text data $\mathcal{D}_r = \{\boldsymbol{y}_j\}_{j=1}^M$; $M \gg N$
**Procedure** TRAIN($\mathcal{D}_p, \mathcal{D}_r$)**:**
1:    Update $\phi_z, \phi_c, \theta, \eta$ by gradient descent on $\mathcal{L}_{\text{ELBO}_p} + \mathcal{L}_{\text{MI}} + \mathcal{L}_{\text{pt}} + \mathcal{L}_{\text{pc}}$
2:    Update $\phi_z, \phi_c, \theta$ by gradient descent on $\mathcal{L}_{\text{ELBO}_r} + \mathcal{L}_{\text{MI}}$
3:    Update $\phi_z, \phi_c, \theta, \eta$ by gradient descent on $\mathcal{L}_{tot}$

---

### 3.3 MUTUAL INFORMATION LOSS

As introduced by previous works (Chen et al., 2016; Zhao et al., 2017; 2018), adding mutual information term to ELBO could alleviate KL collapse effectively and improve the quality of variational posterior. Adding mutual information terms directly imposes the association of content and template latent variables with target sentences. Besides, theoretical proof[2] and experimental results show that introducing mutual information bias is necessary in the presence of preserving-template loss $\mathcal{L}_{\text{pt}}(\boldsymbol{x}^p, \boldsymbol{y}^p)$.

As a result, in our work, the following mutual information term is added to objective

$$\mathcal{L}_{\text{MI}}(y) = -I(z, y) - I(c, y).$$

### 3.4 TRAINING PROCESS

The final loss of VTM is made up of the ELBO losses and extra losses:

$$\mathcal{L}_{tot}(x^p, y^p, y^r) = \mathcal{L}_{\text{ELBO}_p}(x^p, y^p) + \mathcal{L}_{\text{ELBO}_r}(y^r) + \lambda_{\text{MI}}(\mathcal{L}_{\text{MI}}(y^p) + \mathcal{L}_{\text{MI}}(y^r))$$
$$+ \lambda_{\text{pt}}\mathcal{L}_{\text{pt}}(x^p, y^p) + \lambda_{\text{pc}}\mathcal{L}_{\text{pc}}(x^p, y^p), \qquad (x^p, y^p) \in \mathcal{D}_p, y^r \in \mathcal{D}_r.$$

$\lambda_{\text{MI}}, \lambda_{\text{pt}}$ and $\lambda_{\text{pc}}$ are hyperparameters with respect to auxiliary losses.

The training procedure is shown in Algorithm 1. The parameters of generation network $\theta$ and posterior network $\phi_{z,c}$ could be trained jointly by both table-text pair data and raw text data. In this way, a large number of raw text data can be used to enrich the generation diversity.

## 4 EXPERIMENT

### 4.1 DATASETS AND BASELINE MODELS

**Dataset.** We perform the experiment on SPNLG (Reed et al., 2018)[3] and WIKI (Lebret et al., 2016; Wang et al., 2018b). Two datasets come from two different domains. The former is a collection of restaurant descriptions, which expands the E2E dataset[4] into a total of $204,955$ utterances with more varied sentence structures and instances. The latter contains $728,321$ sentences of biographies from Wikipedia. To simulate the environment that a large number of raw texts provided, we just use part of the table-text pairs from two datasets, leaving most of the instances as raw texts. Concretely, for two datasets, we initially keep the ratio of table-text pairs to raw texts as 1:10. For WIKI dataset, in addition to the data from *WikiBio* (Lebret et al., 2016), the raw text data is further extended by the biographical descriptions of people[5] from external *Wikipedia Person and Animal* Dataset (Wang et al., 2018a). The statistics for the number of table-text pairs and raw texts in the training, validation and test sets are shown in Table 2.

**Evaluation Metrics.** For WIKI dataset, we evaluate the generation quality based on BLEU-4, NIST, ROUGE-L (F-score). For SPNLG, we use BLEU-4, NIST, METEOR, ROUGE-L (F-score), and CIDEr. We use the same automatic evaluation script from E2E NLG Challenge[6]. The diversity of generation is evaluated by self-BLEU (Zhu et al., 2018). The lower self-BLEU, the more diversely the model generates.

---

[2] Proof can be found in Appendix C

[3] https://nlds.soe.ucsc.edu/sentence-planning-NLG

[4] http://www.macs.hw.ac.uk/InteractionLab/E2E/

[5] https://eaglew.github.io/patents/

[6] https://github.com/tuetschek/e2e-metrics

| | Train | | Valid | | Test |
|---|---|---|---|---|---|
| Dataset | #table-text pair | #raw text | #table-text pair | #raw text | #table-text pair |
| SPNLG | $14,906$ | $149,058$ | $20,495$ | / | $20,496$ |
| WIKI | $84,150$ | $841,507$ | $72,831$ | $42,874$ | $72,831$ |

Table 2: Dataset statistics in our experiments.

**Baseline models.** We implement the following models as baselines:

- **Table2seq**: Table2seq model first encodes the table into hidden representations then generates the sentence in a sequence-to-sequence architecture (Sutskever et al., 2014). For a fair comparison, we apply the same table-encoder architecture as in Section 2 and the same LSTM decoder with attention mechanism as our model. The model is only trained on pair-wise data. During the testing, we generate five sentences with beam size ranging from one to five to increase some variations. We denote the model as **Table2seq-beam**. We also implement the decoding with forward sampling strategy (namely **Table2seq-sample**). Moreover, to incorporate raw data, we first pretrain the decoder using raw text as a language model, then train Table2seq on the table-text pairs, which is noted as **Table2seq-pretrain**. Table2seq-pretrain has the same decoding strategy as Table2seq-beam.

- **Temp-KN**: Template-KN model (Lebret et al., 2016) first generates a template according to the interpolated 5-gram Kneser-Ney (KN) language modeled over sentence templates, then replaces the special token for the field with the corresponding words from the table.

The hype-parameters of the VTM are chosen based on the lowest $\mathcal{L}_{\mathrm{ELBO}_p}$ on the validation set of SPNLG and $\mathcal{L}_{\mathrm{ELBO}_p} + \mathcal{L}_{\mathrm{ELBO}_r}$ on the validation set of WIKI. Word embeddings are randomly initialized with 300-dimension. During training, we use Adam optimizer (Kingma & Ba, 2015) with the initial learning rate as 0.001. Details on hyperparameters are listed in Appendix D.

## 4.2 EXPERIMENTAL RESULTS ON SPNLG DATASET

**Quantitative analysis.** According to the results in Table 3, we find that our variational template machine (VTM) can generally produce sentences with more diversity under a promising performance in terms of BLEU metrics. Table2seq with beam search algorithm (Table2seq-beam), which is only trained on parallel data, generates the most fluent sentences, but its diversity is rather poor. Although the sampling decoder (Table2seq-sample) gets the lowest self-BLEU, it sacrifices the fluency at the cost. Table2seq performs even worse when the decoder is pre-trained by raw data as a language model. Because there is still a gap between the language model and data-to-text task, the decoder fails to learn how to use raw text in the generation of data-to-text stage. On the contrary, VTM can make full use of the raw data with the help of content variables. As a template-based model, Temp-KN receives the lowest self-BLEU score, but it fails to generate fluent sentences.

**Ablation study.** To study the effectiveness of the auxiliary loses and the augmented raw texts, we progressively remove the auxiliary losses and raw data in the ablation study. We reach the conclusions as follows.

- Without the preserving-content loss $\mathcal{L}_{\mathrm{pc}}$, the model has a relative decline in generation quality. This implies that, by training the same inference model of content variable in pairwise data, preserving-content loss provides an effective instruction for learning the content space.

- VTM-noraw is the model trained without using raw data, where only the loss functions in Section 3.1 are optimized. Comparing with VTM-noraw, VTM gets a substantial improvement in generation quality. More importantly, without extra raw text data, there is also a decline in diversity (self-BLEU). Experimental results show that raw data plays a valuable role in improving both generation quality and diversity, which is often neglected by previous studies.

- We further remove the mutual information loss and preserving-template loss from VTM-noraw model. Both generation quality and diversity continuously decline, which verifies the effectiveness of the two losses. Moreover, the automatic evaluation results of VTM-noraw-$\mathcal{L}_{\mathrm{MI}}$-$\mathcal{L}_{\mathrm{pt}}$ empirically show that preserving-template loss may be a hinder if we only add it during the training, as illustrated in Section 3.3.

| Methods | BLEU | NIST | METEOR | ROUGE | CIDEr | Self-BLEU |
|---------|------|------|--------|-------|-------|-----------|
| Table2seq-beam | 40.61 | 6.31 | 38.67 | 56.95 | 3.74 | 97.14 |
| Table2seq-sample | 34.97 | 5.68 | 35.46 | 52.74 | 3.00 | 65.69 |
| Table2seq-pretrain | 40.56 | 6.33 | 38.51 | 56.32 | 3.75 | 100.00 |
| Temp-KN | 6.45 | 0.45 | 12.53 | 27.60 | 0.23 | 37.85 |
| **VTM** | 40.04 | 6.25 | 38.31 | 56.48 | 3.64 | 88.77 |
| -$\mathcal{L}_{pc}$ | 39.58 | 6.24 | 38.30 | 56.24 | 3.69 | 87.20 |
| VTM-noraw | 39.94 | 6.22 | 38.42 | 56.72 | 3.66 | 88.92 |
| -$\mathcal{L}_{\mathrm{MI}}$ | 38.33 | 6.02 | 37.77 | 55.92 | 3.51 | 96.55 |
| -$\mathcal{L}_{\mathrm{MI}}$-$\mathcal{L}_{\mathrm{pt}}$ | 39.63 | 6.24 | 38.35 | 56.36 | 3.70 | 92.54 |

Table 3: Result for SPNLG data set. Under the 0.05 significance level, VTM gets significantly higher results in all the fluency metrics than all the baselines except Table2seq-beam.

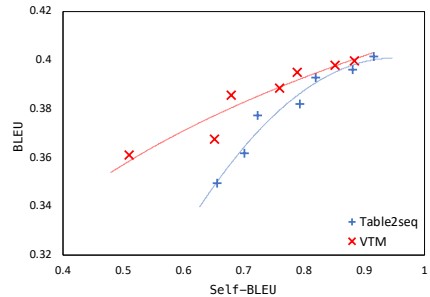

Figure 3: Quality-diversity trade-off curve on SPNLG dataset.

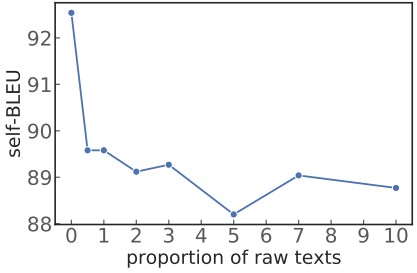

Figure 4: Self-BLEU and the proportion of raw texts to table-sentence pairs.

**Experiment on quality and diversity trade-off.** The quality and diversity trade-off is further analyzed to illustrate the superiority of VTM. In order to evaluate the quality and diversity under different sampling methods, we conduct experiment on sampling from the softmax with different temperatures. Sampling from the softmax with *temperature* is commonly applied to shape the distribution (Ficler & Goldberg, 2017; Holtzman et al., 2019). Given the logits $u_{1:|V|}$ and temperature $\tau$, we sample from the distribution:

$$p(y_t = V_l | y_{<t}, x, z, \tau) = \frac{\exp\left(u_l/\tau\right)}{\sum_{l'} \exp\left(u_{l'}/\tau\right)}$$

When $\tau \to 0$, it approaches greedy decoding. When $\tau = 1.0$, it is the same as forward sampling. In the experiment, we gradually adjust temperature from 0 to 1, taking $\tau = 0.1, 0.2, 0.3, 0.5, 0.6, 0.9, 1.0$. BLEU and self-BLEU under different temperatures are evaluated for both Table2seq and VTM. The self-BLEU in different temperatures and BLEU and self-BLEU curves are plotted in Figure 3. It empirically demonstrates the trade-off between the generation quality and diversity. By sampling from different temperatures, we can plot the portfolios of (Self-BLEU, BLEU) pairs of Table2seq and VTM. The closer the curve is to the upper left, the better the performance of the model. VTM generally gets lower self-BLEU with more diverse outputs under the comparable level of BLEU score.

**Human evaluation** In addition to the quantitative experiments, human evaluation is conducted as well. We randomly select 120 generated samples (each has five sentences) and ask three annotators to rate them on a 1-5 Likert scale in terms of the following features:

- **Accuracy**: whether the generated sentences are consistent with the content in the table.

- **Coherence**: whether the generated sentences are coherent.

- **Diversity**: whether the sentences have as many patterns/structures as possible.

Based on the qualitative results in Table 4, VTM generates the best sentences with the highest accuracy and coherence. Besides, VTM is able to obtain the comparable diversity with Table2seq-sample and Temp-KN. Compared with the model without using raw data (VTM-no raw), there is a significant improvement in diversity, which indicates that raw data essentially enriches the latent

| Methods | Accuracy | Coherence | Diversity |
|---|---|---|---|
| Table2seq-sample | 3.44 | 4.54 | **4.87** |
| Temp-KN | 2.90 | 2.78 | **4.85** |
| VTM | **4.44** | **4.84** | 4.33 |
| VTM-noraw | 4.33 | 4.62 | 3.44 |

Table 4: Human evaluation results on different models. The bold numbers are significantly higher then others under 0.01 significance level.

| Methods | BLEU | NIST | ROUGE | Self-BLEU |
|---|---|---|---|---|
| Table2seq-beam | 26.74 | 5.97 | 48.20 | 92.00 |
| Table2seq-sample | 21.75 | 5.32 | 42.09 | 36.07 |
| Table2seq-pretrain | 25.43 | 5.44 | 45.86 | 99.88 |
| Temp-KN | 11.68 | 2.04 | 40.54 | 73.14 |
| **VTM** | 25.22 | 5.96 | 45.36 | 74.86 |
| -$\mathcal{L}_{pc}$ | 22.16 | 4.28 | 40.91 | 80.39 |
| VTM-noraw | 21.59 | 5.02 | 39.07 | 78.19 |
| -$\mathcal{L}_{\mathrm{MI}}$ | 21.30 | 4.73 | 40.99 | 79.45 |
| -$\mathcal{L}_{\mathrm{MI}}$-$\mathcal{L}_{\mathrm{pt}}$ | 16.20 | 3.81 | 38.04 | 84.45 |

Table 5: Results for WIKI dataset. All the metrics are significant under 0.05 significance level.

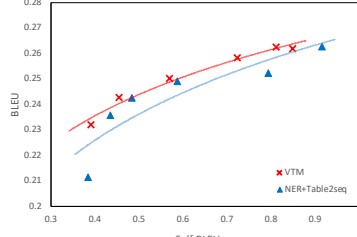

Figure 5: Quality-diversity trade-off curve compared with NER+Table2seq.

template space. Although obtaining the highest scores in diversity for Table2seq-sample and Temp-KN, their generation qualities are much inferior to the VTM, and comparable generation quality is the prerequisite when comparing the diversity.

**Experiment on the diversity under different proportions of raw.** In order to show how much raw data may contribute to the VTM model, we train the model under different proportions of raw data to pairwise data in training. Specifically, we control the ratio of raw sentences to the table-text pairs under 0.5:1, 1:1, 2:1, 3:1, 5:1, 7:1 and 10:1. As shown in Figure 4, the self-BLEU rapidly decreases even adding a small number of raw data, and continuously decreases until the ratio equals 5:1. The improvement is marginal after adding more than 5 times of raw data.

**Case study.** According to Table 8 (in Appendix E), despite template-like structures vary much in a forward sampling model, the information in sentences may be wrong. For example, Sentence 3 says that the restaurant is a Japanese place. Notably, VTM produces correct texts with more diversity of templates. VTM is able to generate different number of sentences and conjunctions. For example, "[*name*] is a [*food*] place in [*area*] with a price range of [*priceRange*]. It is a [*eatType*]." (Sentence 1, two sentences, "with" aggregation), "[*name*] is a [*eatType*] with a price range of [*priceRange*]. It is in [*area*]. It is a [*food*] place." (Sentence 2, three sentences, "with" aggregation), "[*name*] is a [*food*] restaurant in [*area*] and it is a [*food*]." (Sentence 4, one sentence, "and" aggregation).

### 4.3 EXPERIMENTAL RESULTS ON WIKI DATASET

Table 5 shows the results for WIKI dataset, the same conclusions can be drawn as in the results in SPNLG dataset for both the **quantitative analysis** and **ablation study**. VTM is able to generate sentences with the comparable quality as Table2seq-beam but more diversity.

**Comparison with the pseudo-table-based method.** Another way to incorporate raw data is to construct pseudo-table from the given sentence by applying a sentence-to-table backward model via name entity recognition (NER). However, when the type of entities is complicated, such as in product introduction, or the raw data comes from the different domains as pairwise data, the commonly-used model for NER cannot provide accurate pseudo-tables. In this experiment, we replace 841,507 biography raw sentences with 101,807 sentences that describe the animals (Wang et al., 2018b) to test the generalization of our model in raw data of different domains. **NER+Table2seq** is the two-step model that first constructs the pseudo-table by a Bi-LSTM-CRF (Huang et al., 2015) model trained from the table-text pairs, then trains Table2seq from both table-text pairs and pseudo-table-text pairs. We control the temperature in decoding method as previous, and results are plotted in Figure 5. We find that compared with NER+Table2seq, the curve of VTM is closer to the upper left,

|       | Table2seq | VTM-noraw | VTM |
|-------|-----------|-----------|-----|
| Train | ∼30min / 6 epochs | ∼30min / 6 epochs | ∼160min / 15 epochs |
| Test  | ∼80min | ∼80min | ∼80min |

Table 6: Computational cost for each model.

| | |
|---|---|
| Table | **name**[Jack Ryder], **country**[Australia], **fullname**[John Ryder], **nickname**[the king of Collingwood], **birth_date**[8 August 1889], **birth_place**[Collingwood, Victoria, Australia], **death_date**[4 April 1977], **death_place**[Fitzroy, Victoria, Australia], **club**[Victoria], **testdebutyear**[1920 england], **aritcle_title**[Jack Ryder (cricketer)] |
| Reference | John "Jack" Ryder, mbe (8 August 1889 – 3 April 1977) was a cricketer who played for Victoria and Australia. |
| Table2seq-sample | **1:** john Ryder ( 8 August 1889 – 3 April 1977) was an Australian cricketer . 
 **2:** john Ryder Ryder ( 8 August 1889 – 3 April 1977) was an Australian cricketer . 
 **3:** john Ryder Ryder ( 8 August 1889 – 3 April 1977) was an Australian cricketer who played for glouces-tershire cricket club in 1912 . 
 **4:** john Ryder ( 8 August 1889 – 3 April 1977) was an Australian cricketer . 
 **5:** john Ryder oliveira ( 8 August 1889 – 3 April 1977) was an Australian test cricketer who played against great Britain with international cricket club . |
| Temp-KN | **1:** jack Ryder ( born August 8, 1889) is a former professional cricketer) . 
 **2:** "jack" Ryder ( born August 8, 1889) is a former professional cricketer) who played in the national football league. 
 **3:** jack Ryder ( born 8 August 1889 in Collingwood, Victoria,) is a former professional cricketer) . 
 **4:** Jack Ryder ( born August 8, 1889, in Collingwood, Victoria, Australia) is a former professional football player who is currently a member of the united states . 
 **5:** jack Ryder ( born August 8, 1889) is a former professional cricketer) . |
| VTM-noraw | **1:** John Ryder (8 August 1889 – 4 April 1977) was an Australian cricketer. 
 **2:** Jack Ryder (born August 21, 1951 in Melbourne, Victoria) was an Australian cricketer. 
 **3:** John Ryder (21 August 1889 – 4 April 1977) was an Australian cricketer. 
 **4:** Jack Ryder (8 March 1889 – 3 April 1977) was an Australian cricketer. 
 **5:** John Ryder (August 1889 – April 1977) was an Australian cricketer. |
| VTM | **1:** John Ryder (8 August 1889 – 4 April 1977) was an Australian cricketer. 
 **2:** John Ryder (born 8 August 1889) was an Australian cricketer. 
 **3:** Jack Ryder (born August 9, 1889 in Victoria, Australia) was an Australian cricketer. 
 **4:** John Ryder (August 8, 1889 – April 4, 1977) was an Australian rules footballer who played for Victoria in the Victorian football league (VFL). 
 **5:** John Ryder, also known as the king of Collingwood (8 August 1889 – 4 April 1977) was an Australian cricketer. |

Table 7: An example of the generated text by our model and the baselines on WIKI dataset.

which implies that VTM can generate more diverse (lower Self-BLEU) under the commensurate BLEU.

**Computational cost.** We further compare the computational cost of VTM with other models, for both training and testing phases. We train and test the models on a single Tesla V100 GPU. The time spent to reach the lowest ELBO in the validation set is listed in Table 6. VTM is trained about five times longer than the baseline Table2seq model (160 minutes, 15 epochs in total) because of the training of an extra large number of raw data (84k pairwise data and 841k raw texts). In the testing phase, VTM enjoys the same speed as other competitor models, approximately 80 minutes to generate 72k wiki sentences in the test set.

**Case study.** Table 7 shows an example of sentences generated by different models. Although forward sampling enables the Table2seq model to generate diversely, it is more likely to generate incorrect and irrelevant content. For example, it generates the wrong club name in Sentence 3. By sampling from template space, VTM-noraw can generate texts with multiple templates, like different expressions for birth date and death date, while preserving readability. Furthermore, with extra raw data, VTM is able to generate more diverse expressions, which other models cannot produce, such as "[*fullname*], also known as [*nickname*] ([*birth_date*] – [*daeth_date*]) was a [*country*] [*article_name_4*]." (Sentence 5). It implies that raw sentences not in the pairwise dataset could additionally enrich the information in template space.

## 5 RELATED WORK

**Data-to-text Generation.** Data-to-text generation aims to produce summary for the factual structured data, such as numerical table. Neural language models have made distinguished progress by generating sentences from the table in an end-to-end style. Jain et al. (2018) proposed a mixed hierarchical attention model to generate weather report from the standard table. Gong et al. (2019) proposed a hierarchical table-encoder and a decoder with dual attention. Although encoder-decoder models can generate fluent sentences, they are criticized for deficiency in sentence diversity. Other works focused on controllable and interpretable generation by introducing templates as latent variables. Wiseman et al. (2018) designed a Semi-HMM decoder to learn discrete templates representation, and Dou et al. (2018) created a platform, Data2TextStudio, equipped with a Semi-HMMs model, to extract template and generate from table input in an interactive way.

**Semi-supervised Learning From Raw Data.** It is easier to acquire raw text than to get structured data, and most neural generators cannot make the best use of raw text, universally. Ma et al. (2019) proposed that encoder-decoder framework may fail when not enough parallel corpus is provided. In the area of machine translation, back-translation have been proved to be an effective method to utilize monolingual data (Sennrich et al., 2016; Burlot & Yvon, 2018).

**Latent Variable Generative Model.** Deep generative models, especially variational autoencoders (VAE) (Kingma & Welling, 2014) have shown a promising performance in generation. Bowman et al. (2016) showed that a RNN-based VAE model can produce diverse and well-formed sentences by sampling from the prior of continuous latent variable. Recent works explored methods to learn disentangled latent variables (Hu et al., 2017a; Zhou & Neubig, 2017; Bao et al., 2019). For instance, Bao et al. (2019) devised multi-task losses adversarial losses to disentangle the latent space into syntactic space and semantic space. Motivated by the idea of back-translation and variational autoencoders, VTM model proposed in this work can not only fully utilize the non-parallel text corpus, but also learn a disentangled representation for template and content.

## 6 CONCLUSION

In this paper, we propose the Variational Template Machine (VTM) based on a semi-supervised learning approach in the VAE framework. Our method not only builds independent latent spaces for template and content for diverse generation, but also exploits raw texts without tables to further expand the template diversity. Experimental results on two datasets show that VTM outperforms the model without using raw data in terms of both generation quality and diversity, and it can achieve a comparable quality in generation with Table2seq, as well as promote the diversity by a large margin.

### ACKNOWLEDGMENTS

We thank the anonymous reviewers for their insightful comments. Hao Zhou and Zhongyu Wei are the corresponding authors of this paper.

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

## A    EXPLANATION FOR PRESERVING-CONTENT LOSS

The first term of $-\mathcal{L}_{\mathrm{pc}}(x, y)$ is equivalent to:

$$
\begin{aligned}
\mathbb{E}_{q_c(c|x)}||c - h||^2 &= \mathbb{E}_{q_c(c|x)} \sum_{i=1}^{K}(c_i - h_i)^2 \\
&= \sum_{i=1}^{K} \mathbb{E}_{q_c(c|x)}(c_i - h_i)^2 \\
&= \sum_{i=1}^{K}[(\mathbb{E}(c_i - h_i))^2 + \mathrm{var}(c_i)] \\
&= \sum_{i=1}^{K}[(E(c_i) - h_i)^2 + \mathrm{var}(c_i)] \\
&= \sum_{i=1}^{K}[(\mu_i - h_i)^2 + \Sigma_{ii}] \\
&= ||\mu - h||^2 + tr(\Sigma)
\end{aligned}
$$

When we minimize it, we jointly minimize the distance between mean of approximated posterior distribution, and the trace of the co-variance matrix.

## B    PROOF FOR ANTI-INFORMATION PROPERTY OF ELBO

Consider the KL divergence over the whole dataset (or a mini-batch of data), we have

$$
\begin{aligned}
\mathbb{E}_{x \sim p(x)}[D_{\mathrm{KL}}(q(z|x)\|p(x))] &= \mathbb{E}_{q(z|x)p(x)}[\log q(z|x) - \log p(z)] \\
&= -H(z|x) - \mathbb{E}_{q(z)} \log p(z) \\
&= -H(z|x) + H(z) + D_{\mathrm{KL}}(q(z)\|p(z)) \\
&= I(z, x) + D_{\mathrm{KL}}(q(z)\|p(z))
\end{aligned}
$$

where $q(z) = \mathbb{E}_{x \sim \mathcal{D}}(q(z|x))$ and $I(z, x) = H(z) - H(z|x)$. Since KL divergence can be viewed as a regularization term in ELBO loss, When ELBO is maximized, the KL term is minimized, and mutual information between $x$ and latent $z$, $I(z, x)$ is minimized. This implies that $z$ and $x$ eventually become more independent.

## C    PROOF FOR THE PRESERVING-TEMPLATE LOSS WHEN POSTERIOR COLLAPSE HAPPENS

When posterior collapse happens, $D_{\mathrm{KL}}(q(z|y)\|p(z)) \approx 0$,

$$
\begin{aligned}
\mathcal{L}_{pt}(Y, \tilde{Y}) &= \mathbb{E}_{\tilde{y} \sim p(\tilde{y}), y \sim p(y)} \mathbb{E}_{z \sim q(z|y)} \log p_\eta(\tilde{y}|z) \\
&= \mathbb{E}_{\tilde{y} \sim p(\tilde{y})} \mathbb{E}_{z \sim p(z)} \log p_\eta(\tilde{y}|z) \\
&= \int_{\tilde{y}} p(\tilde{y}) \int_z p(z) \log p_\eta(\tilde{y}|z) \mathrm{dz}\, \mathrm{d}\tilde{y} \\
&= \int_z p(z) \int_{\tilde{y}} p(\tilde{y}) \log p_\eta(\tilde{y}|z) \mathrm{dz}\, \mathrm{d}\tilde{y} \\
&= \mathbb{E}_z \mathbb{E}_{\tilde{y}}[\log p_\eta(y)|z] = \mathbb{E}_{\tilde{y}} \log p_\eta(y)
\end{aligned}
$$

During the back-propagation,

$$
|| \bigtriangledown_z \mathcal{L}_{pt}(Y, \tilde{Y})|| = 0
$$

thus, $\phi_z$ is not updated.

## D   IMPLEMENTATION DETAILS

For the model trained on WIKI dataset, the the dimension of latent template variable is set as 100, and the dimension of latent content variable is set as 200. The dimension of the hidden for table is 300. For the hyperparameters of total loss $\mathcal{L}_{tot}$, we set $\lambda_{\mathrm{MI}} = 0.5$, $\lambda_{\mathrm{pt}} = 1.0$ and $\lambda_{\mathrm{pc}} = 0.5$.

For the model trained on SPNLG dataset, the dimension of latent template variable is set as 64, and the dimension of latent content variable is set as 100. The dimension of the hidden for table is also 300. For the hyperparameters of total loss $\mathcal{L}_{tot}$, we set $\lambda_{\mathrm{MI}} = \lambda_{\mathrm{pt}} = \lambda_{\mathrm{pc}} = 1.0$.

## E   CASE STUDY ON SPNLG EXPERIMENT

| Table | **name**[nameVariable], **eatType**[pub], **food**[French], **priceRange**[20-25], **area**[riverside] |
|---|---|
| Reference | nameVariable is a French place with a price range of 20-25. It is in riverside. It is a pub. |
| Table2seq-sample | **1:** nameVariable is a pub with a price range of 20-25. It is a French restaurant in riverside.
**2:** nameVariable is a French restaurant in riverside with a price range of 20-25. nameVariable is a pub.
**3:** nameVariable is a pub with a price range of 20-25 and nameVariable is a French restaurant in riverside.
**4:** nameVariable is a pub with a price range of 20-25, also it is in riverside. it is a Japanese place.
**5:** nameVariable is a pub with a average rating and it is a French place in riverside. |
| Temp-KN | **1:** nameVariable is in riverside, also it is in riverside.
**2:** nameVariable is a French restaurant.
**3:** nameVariable is the best restaurant.
**4:** nameVariable is in riverside, and nameVariable is in [location].
**5:** nameVariable is in. Its a French restaurant and it is in [location] with food and, even if nameVariable is [food_qual], it is the best place. |
| VTM-noraw | **1:** nameVariable is a pub with a price range of 20-25. It is a French place in riverside.
**2:** nameVariable is a pub with a price range of 20-25. it is a pub. It is in riverside.
**3:** nameVariable is a French place in riverside with a price range of 20-25. It is a pub.
**4:** nameVariable is a French place in riverside with a price range of 20-25. It is a pub.
**5:** nameVariable is a French place in riverside with a price range of 20-25. It is a pub. |
| VTM | **1:** nameVariable is a French place in riverside with a price range of 20-25. It is a pub.
**2:** nameVariable is a pub with a price range of 20-25. It is in riverside. It is a French place.
**3:** nameVariable is a French pub in riverside with a price range of 20-25, and it is a pub.
**4:** nameVariable is a French restaurant in riverside and it is a pub.
**5:** nameVariable is a French place in riverside with a price range of 20-25. It is a pub. |

Table 8: An example of the generated text by our model and the baselines on SPNLG dataset.

