# OpenReview forum: "Variational Template Machine for Data-to-Text Generation"
_ICLR.cc/2020/Conference — Accept (Poster)_

### Official Review · AnonReviewer1 · 2019-10-08
**Official Blind Review #1**

**Rating:** 8

**Review:**

The paper is interesting and proposes a novel approach for addressing a currently not largely considered problem.
The proposed model is sound and appropriate, as it relies on state-of-the-art methodological arguments.
The derivations are correct; this concerns both the model definition and the algorithmic derivations of model training and inference.
The experimental evaluation is adequate: it compares to many popular approaches and on several datasets; the outcomes are convincing.
It would be good if the authors could provide an analysis of the computational costs of their methods, as well as of the considered competitors.

**Experience Assessment:**

I have published in this field for several years.

**Review Assessment: Checking Correctness Of Derivations And Theory:**

I carefully checked the derivations and theory.

**Review Assessment: Checking Correctness Of Experiments:**

I carefully checked the experiments.

**Review Assessment: Thoroughness In Paper Reading:**

I read the paper thoroughly.

---

> ### Author Response · Authors · 2019-11-15
> **Response to Review #1**
>
> Thanks very much for your valuable comments.
>
> Q: It would be good if the authors could provide an analysis of the computational costs of their methods, as well as of the considered competitors.
>
> A: We compare the training and testing time cost on the WIKI dataset, and with raw data added, VTM spends more time on training but the same time on generation as Table2seq. Here is the detailed time spent ( train and test on single Tesla V100 GPU), for test computational cost, we record how much time to generate 72k sentences.
>
>                 |          Table2seq              |       VTM-noraw               |              VTM
> --------------------------------------------------------------------------------------------------------------------
> Train        |  ～30 mins (6 epochs)  | ～30 mins (6 epochs)  |  ～160 mins (15 epochs)
> --------------------------------------------------------------------------------------------------------------------
> Test         |          ~80min                  |         ~80min                     |           ~80min
>
> VTM gives the same speed for generating sentences, but it takes more time for training, which are cost to learn the large-scaled unlabeled data, and is acceptable.
>
> Additionally, we've added some new experiments with more sophisticated setups. In the experiments (result see Section 4.3, Figure 3), we control the same decoding strategy under the same temperature, and plot their BLEU and Self-BLEU scores in Figure 3 to analyze the quality-diversity trade-off. Experimental results show that compared to Table2seq, VTM always gives better self-BLEU when they have the same BLEU, and gives better BLEU under the same Self-BLEU. This shows that VTM outperforms Table2text consistently.

---

### Official Review · AnonReviewer3 · 2019-10-19
**Official Blind Review #3**

**Rating:** 3

**Review:**

This paper proposes Variational Template Machine (VTM), a generative model to generate textual descriptions from structured data (i.e., tables). VTM is derived from the variational autoencoder, where the input is a row entry from a table and the output is the text associated with this entry. The authors introduce two latent variables to model contents and templates. The content variable is conditioned on the table entry, and generates the textual output together with the template variable. The model is trained on both paired table-to-text examples as well as unpaired (text only) examples. Experiments on the Wiki and SpNLG datasets show that models generate diverse sentences, and the overall performance in terms of BLEU is only slightly below the best baseline Table2Seq model that does not generate diverse sentences. The results also show that additional losses for preserving contents and templates introduced by the authors play an important role in the overall model performance.

I have several questions regarding the experiments:
- For the Table2Seq baseline, how was the beam size chosen? Did it have any effect on the performance of the baseline model?
- Did the authors try other sampling methods for Table2Seq? (e.g., top-K or nucleus sampling)
- VTM is only able to achieve comparable performance to Table2Seq in terms of BLEU after including the unlabeled corpus, especially on the Wiki dataset. A way to incorporate this unlabeled data to Table2Seq is by first pretraining the LSTM generator on it before training it on pairwise data (or in parallel). How would this baseline model perform in comparison to VTM?
- In the conclusion section, the authors mentioned that VTM outperforms VAE both in terms of diversity and generation quality. What does this VAE model refer to? The experiments show that VTM is comparable to Table2Seq in terms of quality and is better in terms of diversity.

Generating text from structured data is an interesting research area. However, I am not convinced that the proposed method is a significant development based on the results presented in the paper. There are also many grammatical errors in the paper (e.g., ... only enable to sample in the latent space ..., and many others), so I think the writing of the paper can be improved.

**Experience Assessment:**

I have published in this field for several years.

**Review Assessment: Checking Correctness Of Derivations And Theory:**

I assessed the sensibility of the derivations and theory.

**Review Assessment: Checking Correctness Of Experiments:**

I carefully checked the experiments.

**Review Assessment: Thoroughness In Paper Reading:**

I read the paper at least twice and used my best judgement in assessing the paper.

---

> ### Author Response · Authors · 2019-11-15
> **Response to Review #3**
>
> Thanks a lot for your insightful comments. In the following parts, we will response to your questions one by one.
>
> Q: Quality-diversity trade-off.
> A:
> Quality and diversity is a trade-off in the text generation. As we can find in Table 3 and 6, beam search (or greedy) always receives a higher quality and low diversity whereas forward sampling method can diversify the generation but with a relatively lower quality. Decoding strategy largely interferes our judgment. A fairer comparison is to keep the same decoding strategy then comparing the quality and diversity.
>
> Therefore, we add extensive experiments in Section 4.3 (results refer Figure 4), by applying same decoding method -- sampling under the same softmax temperature, we plot their BLEU scores and Self-BLEU scores in Figure 3, which shows that compared to Table2seq, VTM always gives better self-BLEU when they have the same BLEU, and gives better BLEU under the same Self-BLEU. This shows that VTM outperforms Table2text consistently.
>
> ========
>
> Q: Did the authors try other sampling methods for Table2Seq? (e.g., top-K or nucleus sampling)
> A:
> Thanks very much for your kind notes, We have updated the paper, adding extensive experiments on other sampling methods and human evaluation. Results show that our method consistently outperforms the baseline model (Table2Seq).
>
> Firstly, we sample from the softmax with different temperatures (from 0.1 to 1.0) and plot the BLEU and self-BLEU trade-off curve in Figure 3. The trade-off curves show that compared to Table2seq, VTM always give better self-BLEU when they have the same BLEU, and gives better BLEU under the same Self-BLEU. This shows that VTM outperforms Table2text consistently.
>
> Secondly, human evaluation also shows that VTM can generate sentences with better accuracy and coherence. Besides, comparing to the model without raw data (VTM-no raw), there is a significantly large improvement in diversity.
> Although Table2seq with forward sampling has the highest diversity rating, its quality much worse than VTM. High quality is meaningful when the output quality is good enough.
>
> ========
> Q: A way to incorporate this unlabeled data to Table2Seq is by first pretraining the LSTM generator on it before training it on pairwise data (or in parallel). How would this baseline model perform in comparison to VTM?
> A:
> Yes, pretraining the decoder with large-scaled unlabeled data can be another alternative for including the effectiveness of the large scaled unlabeled data. However, we did a quick run and experimental results in Table 3 and Table 6 show that directly applying decoder pretraining does not get performance gain (even worse than the baseline), which might be caused by the gap between pretraining the generator as a language model and the data-to-text task.
> Here we list the BLEU and self-BLEU scores:
>
> Dataset |           Model             |      BLEU    |   Self-BLEU
> -----------------------------------------------------------------------------
> WIKI      |  Table2seq-beam    |     26.74     |     92.00
>                | Table2seq-pretrain|     25.43     |     99.88
>                |            VTM               |     25.22     |     74.86
> -----------------------------------------------------------------------------
> SPNLG   | Table2seq-beam      |     40.61    |     97.14
>                | Table2seq-pretrain  |     40.56    |    100.00
>                |            VTM                 |      40.04   |     88.77
>
> ========
>
> Q: In the conclusion section, what does this VAE model refer to?
> A: Sorry for misleading, VAE model refers to the VTM without using raw data (i.e. VTM-noraw). We've fixed it in the updated version.
>
> =======
>
> Q: I am not convinced that the proposed method is a significant development based on the results presented in the paper.
> A:
> As introduced in the quality-diversity trade-off, VTM tends to generate more diverse outputs with the same quality (diversity is important for text generation).
> Our proposed VTM can make full use of the raw data to learn an informative template space, and largely enrich the template of generated sentences, thus boost the diversity. VTM can be also a new approach to include unlabeled data for text generation in the VAE framework.
> To our best knowledge, there is no related work using the similar idea in the data-to-text generation.
>
> =======
>
> Q: There are also many grammatical errors in the paper (e.g., ... only enable to sample in the latent space ..., and many others), so I think the writing of the paper can be improved.
> A: Thanks, we will proof-read carefully and fix typos in the next version.

---

### Official Review · AnonReviewer2 · 2019-10-26
**Official Blind Review #2**

**Rating:** 8

**Review:**

The paper proposes an approach to generate textual descriptions from structured data organized in tables, by using a "variational template machine" (VTM), which is essentially a generative model to separately represent template and content as disentangled latent variables to control the generation.

The contribution is well-written and well-motivated, the model exposition is clear, and the results are convincing. The experiment setup, depth, and breadth are particularly convincing. I see no reason to not accept this paper.

Remarks:
- It should be clearly stated which languages feature in the paper. From what I gather, it's only English. How does the method generalize to other languages? How does it scale with (the lack of) resources?

**Experience Assessment:**

I do not know much about this area.

**Review Assessment: Checking Correctness Of Derivations And Theory:**

I did not assess the derivations or theory.

**Review Assessment: Checking Correctness Of Experiments:**

I assessed the sensibility of the experiments.

**Review Assessment: Thoroughness In Paper Reading:**

I made a quick assessment of this paper.

---

> ### Author Response · Authors · 2019-11-15
> **Response to Review #2**
>
> Thanks very much for your valuable comments.
>
> Q: How does the method generalize to other languages? How does it scale with (the lack of) resources?
> A: Our method could be easily generalized to other languages because no language-specific processing or resources are used. Additionally, our proposed VTM may well fit languages with fewer resources, in which case the VTM model with massive raw data (usually cheap to obtain) may significantly boost the finally performances when labeled data are hard to get.
>
> Additionally, we've added some new experiments with more sophisticated setups. In these experiments (result see Section 4.3, Figure 3), we control the decoding strategy with the same temperature, and plot their BLEU scores and Self-BLEU scores in Figure 3 to analyze the quality-diversity trade-off. Experimental results show that compared to Table2seq, VTM always gives better self-BLEU when they have the same BLEU, and gives better BLEU under the same Self-BLEU. This shows that VTM outperforms Table2text consistently.

---

### Author Response · Authors · 2019-11-15
**We update a new version!**

Hi, all. Thanks for reviewing my paper. We've uploaded a new version of our draft, adding more experiments:
- Experiments on the computational cost of the models. (Table 4, Page 7)
- Experiments on quality-diversity trade-off. (Figure 3, Page 6)
- Human evaluation on generation accuracy, coherence and diversity. (Table 7, Page 9)
 Please take a look！

---

### Decision · Program_Chairs · 2019-12-19

**Decision:**

Accept (Poster)

**Comment:**

The paper addresses the problem of generating descriptions from structured data. In particular a Variational Template Machine  which explicitly disentangles templates from semantic content. They empirically demonstrate that their model performs better than existing methods on different methods.

This paper has received a strong acceptance from two reviewers. In particular, the reviewers have appreciated the novelty and empirical evaluation of the proposed approach. R3 has raised quite a few concerns but I feel they were adequately addressed by the reviewers. Hence, I recommend that the paper be accepted.